# Diatom Red List Species Reveal High Conservation Value and Vulnerability of Mountain Lakes

Stefan Ossyssek *, Andrea Maria Hofmann, Juergen Geist and Uta Raeder

Aquatic Systems Biology Unit, Limnological Research Station Iffeldorf, Technical University of Munich, Hofmark 1-3, 82393 Iffeldorf, Germany; andrea_hofmann@posteo.de (A.M.H.); geist@tum.de (J.G.); uta.raeder@tum.de (U.R.)
* Correspondence: stefan.ossyssek@gmx.de

**Abstract:** Mountain lakes are unique and often isolated freshwater habitats that harbour a rich biotic diversity. This high conservation value may be reflected by diatoms, a group of algae that is known for its reliability as a bioindicator, but which has not been studied extensively in mountain lakes of the northern European Alps. In this study, the conservation value of these lakes was assessed by characterizing the number, share, and abundance of diatom Red List (RL) taxa and their relationship with environmental variables, diatom $\alpha$ and $\beta$ diversity (assemblage uniqueness). For this purpose, linear regression models, generalized linear models, and generalized additive models were fitted and spatial descriptors were included when relevant. Of the 560 diatom taxa identified, 64% were on the RL and half of these were assigned a threat status. As hypothesized, a decreasing share of RL species in sediment and littoral samples at higher trophic levels was reflected by higher total phosphorous content and lower Secchi depth, respectively. Species-rich lakes contained a high number of RL taxa, contrasting our hypothesis of a logarithmic relationship. In turn, RL abundance increased with uniqueness, confirming our initial hypothesis. However, some of the most unique sites were degraded by fish stocking and contained low abundances of RL species. The results demonstrate the importance of oligotrophic mountain lakes as habitats for rare freshwater biota and their vulnerability in light of human impact through cattle herding, tourism, damming, and fish stocking. Additional conservation efforts are urgently needed for mountain lakes that are still underrepresented within legal conservation frameworks. Species richness and uniqueness reflect complementary aspects of RL status and thus should be applied jointly. Uniqueness can indicate both pristine and degraded habitats, so that including information on human impacts facilitates its interpretation.

**Keywords:** rare species; bioindication; diatom diversity; uniqueness; eutrophication; fish stocking; small lakes; top-down control



## 1. Introduction

In light of the global biodiversity crisis, it is crucial to prioritize conservation on a global, regional, and local scale so that limited resources can be used effectively [1–6]. While aquatic biodiversity in freshwater habitats is decreasing rapidly [7], conservation measures lag behind those taken to preserve and restore terrestrial and marine ecosystems [8]. Within the European Union, the "Water Framework Directive" (WFD) that was implemented in 2000 aims to prevent the deterioration and enhance the status of aquatic ecosystems [9]. WFD Annex II provides a non-mandatory minimum size limit for reporting the ecological status of lakes of 50 ha, which is applied by most member states [10]. Accordingly, of the presumed 600,000 lakes and ponds in Europe [11], the ecological status was reported for less than 20,000 lentic water bodies [12]. This low sufficiency in reporting for small lakes can be assumed to be accompanied by a lower degree of restoration measures applied at these sites. However, small lakes are known to have a disproportionately high biodiversity relative to their size [13]. This may be due to their collective higher habitat diversity

compared to a single large lake, as demonstrated for littoral macroinvertebrates [14]. This is especially true for small and isolated mountain lakes, which contribute to the overall biodiversity through their highly adapted biota [15,16]. Due to their remoteness, many of these environmentally unique lakes are still pristine [17]. This makes mountain lakes important sentinels for indirect pressures due to global change [18]. In turn, dramatic consequences for these ecosystems can arise through direct pressures, such as cattle herding [19], hut construction [20], and tourism [21], which often lead to the eutrophication of mountain lakes, while fish stocking disturbs their food webs and internal nutrient recycling [22–26]. These direct pressures may be amplified by climate change [20,27–30], which is particularly pronounced in mountain regions such as the European Alps [31,32]. Therefore, it is important to preserve undisturbed oligotrophic mountain lakes and to restore degraded ones.

Diatoms (class Bacillariophyceae) are considered bioindicators that mirror ambient habitat conditions, including nutrient levels [33], pH [34], and water temperature [35]. They reliably indicate eutrophication [36], acidification [37], and consequences of climate change [28]. Moreover, diatoms constitute a significant portion of algal biodiversity, with a known species richness of 10,000–12,000 [38]. Projections of the actual diversity range from 20,000 [39] to 200,000 [40]. This known yet still hidden diversity, along with their indicator function, makes diatoms ideal conservation targets within aquatic ecosystems.

The recently updated Red List for diatoms in Germany [41] can be a valuable reference to assess the conservation value of mountain lakes. The threat status of RL species incorporates their spatial restrictedness, rarity, as well as long- and short-term population trends, and thus the likelihood of their local extinction. On a continental scale, rare taxa may simply be captured by conserving species-rich regions [42], whereas it may be more important to preserve high β diversity to prevent species extinction on a regional scale [43,44]. A high restrictedness and rarity of a taxon should lead to a high contribution to the uniqueness of a site, which in turn reflects its local contribution to β diversity [45]. This is especially important in small lakes with typically high turnover rates, i.e., high β diversity [14]. Consequently, RL species may be effectively conserved by targeting unique assemblages. However, this approach requires the investigation of additional environmental parameters since uniqueness may also reflect degraded habitats [45–47]. Generally, a high number and share of RL diatoms is indicative of rare freshwater habitats [48], which are typically oligotrophic or dystrophic in central Europe [49]. Diatoms can therefore help to identify important refugia for a broader group of threatened freshwater species. Mountain waters are naturally oligotrophic habitats. Accordingly, the application of the diatom RL in the southern European Alps has revealed high shares of RL taxa in mires [50], springs [51], streams [52], and lakes [53]. This has allowed important insights concerning the interaction between diatoms and catchment geology [54,55], water chemistry [56], bryophytes [51], stream flow regime [52,57], and geodiversity [55]. The conservation value remained mostly unknown, especially regarding the updated RL from 2018, for lakes from the northern European Alps. Hence, the aim of this study is (1) to identify the environmental correlates of RL species in this region and (2) to investigate whether α diversity and uniqueness of diatom assemblages reflects the richness, share, and abundance of Red List diatoms (hereafter called "RL indices"). Specifically, we hypothesize that: (1) lake trophic status is negatively correlated with RL indices based on previous findings for springs, mires, and lakes from the southern Alps; (2) a positive logarithmic correlation exists between diatom α diversity and RL species richness. This is based on the assumed increasing diatom species richness with lake trophic level [58–60], which will in turn lead to a lower share of RL taxa (Hypothesis 1). Finally, we hypothesize (3) that the uniqueness of diatom assemblages has a positive correlation with RL indices and is highest in either unimpaired and environmentally unique lakes or in impaired lakes.

## 2. Materials and Methods

### 2.1. Study Site

Most lakes in this study are located on lime bedrock and therefore are well-buffered, as reflected by their pH values of between 8 and 9 (Tables 1 and S1). The altitudinal gradient comprises the vegetation zones of montane forest (750–1400 m asl), subalpine forest (1400–1700 m asl), and alpine meadows (1700–2500 m asl, Figure 1).

Most lakes were formed by cirque glaciers, they are typically small (<3 ha) and shallow (<10 m), consisting of one main basin (Table 1). Two of the lakes are karstic, they are almost round and deeper compared to their surface area than the other lakes. One of these lakes (SieG) can be considered an outlier in terms of water chemistry due to a strong groundwater influence, resulting in a very high conductivity. Depending on the mountain group, different geological settings can be found in the dataset. A rough structuring reveals that the studied lakes in the "Berchtesgadener Alpen", "Chiemgauer Alpen", "Rofangebirge", "Karwendel", and "Mieminger Kette" are mainly on limestone that consists of calcium carbonate ($CaCO_3$), while in the "Lechtaler Alpen" they are on dolomite ($CaMg(CO_3)_2$). In the "Wettersteingebirge", they are partly on dolomite and partly on limestone, and in the "Allgäu" and "Bayerische Voralpen" limestone, dolomite, and marl are often closely interlaced. Seven lakes are stocked with fish according to personal communications of the lake owners or publicly available data (see Supplementary Material). Various lakes are influenced by either intensive cattle herding (traces of trampling and excretion near the lake), tourism, damming, or a combination of these factors based on observations in the field. The relative strength of these influences is not known.

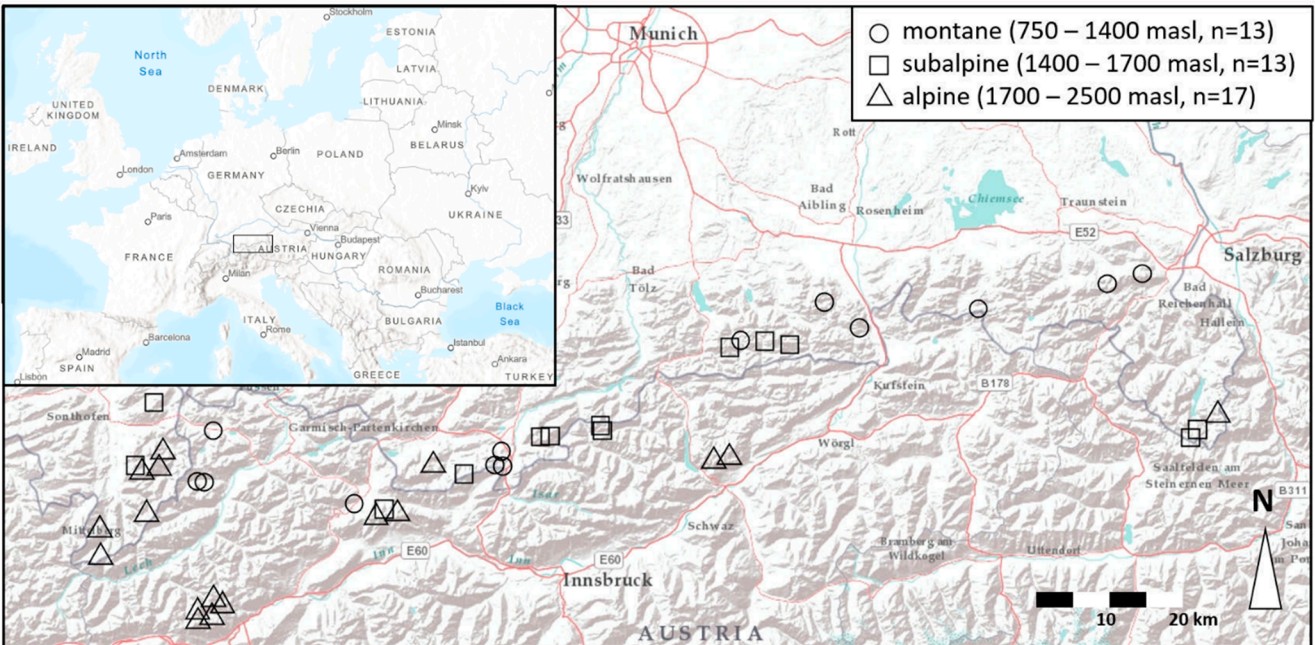

**Figure 1.** The 43 study lakes are spread across the Bavarian and Tirolian Alps in Germany and Austria, spanning a longitudinal gradient of c. 220 km and a latitudinal gradient of c. 50 km.

**Table 1.** Ecologically important morphological, physical, and chemical parameters were assessed for 43 mountain lakes in the northern European Alps ranging from 760 to 2469 m asl. The dashed box indicates an outlier lake that is strongly influenced by groundwater inflow. ABT = August bottom temperature, AST = August surface temperature. For abbreviations of other variables see Table S1. Abbreviations of the main bedrock type ("Geo."): Ls = Limestone, Do = Dolomite, Ma = Marvel, Mi = Mixed. Abbreviations of the mountain regions ("Reg.") are from west to east: A = Allgäu, L = Lechtaler Alpen, M = Mieminger Kette, W = Wettersteingebirge, K = Karwendel, R = Rofangebirge, V = Bayerische Voralpen, C = Chiemgauer Alpen, B = Berchtesgadener Alpen. Human impact is denoted by "+": F = Fish stocked lakes, C = Intensive cattle herding, T = Nearby touristic infrastructure, D = Dammed lakes.

| Lake | Alt. | Area | Depth | Sec. | NO$_3$ | TP | Si | NH$_4$ | Na | Ca | Mg | pH | Cond | O$_2$ | ABT | AST | DIN:TP | Geo. | Reg. | F | C | T | D |
|---|---|---|---|---|---|---|---|---|---|---|---|---|---|---|---|---|---|---|---|---|---|---|---|
| | m asl | ha | m | m | mg/L | µg/L | mg/L | µg/L | mg/L | mg/L | mg/L | | µS/cm | % | °C | °C | | | | | | | |
| Fal | 760 | 1.0 | 15.0 | 7.0 | 1.84 | 9.0 | 0.18 | 17.0 | 2.11 | 42.3 | 6.7 | 8.0 | 266.3 | 96 | 5.8 | 12.8 | 205.4 | Ls | C | | | | |
| Bic | 955 | 1.4 | 11.0 | 4.4 | 0.76 | 4.3 | 0.77 | 32.3 | 0.30 | 49.2 | 14.7 | 7.9 | 360.2 | 61 | 8.4 | 21.7 | 185.7 | Mi | V | + | | | |
| Fri | 973 | 1.2 | 5.5 | 4.6 | 0.93 | 7.2 | 0.15 | 83.7 | 0.48 | 39.6 | 5.4 | 8.6 | 224.3 | 110 | 13.9 | 18.6 | 140.1 | Ls | C | | | | |
| Sut | 995 | 1.4 | 1.5 | 1.5 | 0.67 | 4.8 | 0.76 | 60.8 | 1.62 | 41.2 | 6.7 | 8.0 | 280.7 | 82 | 14.1 | 17.5 | 153.2 | Mi | V | | | | |
| Laut | 1013 | 12.0 | 18.0 | 6.0 | 0.21 | 13.5 | 0.25 | 50.8 | 0.10 | 46.1 | 16.0 | 8.6 | 310.0 | 102 | 8.2 | 16.0 | 19.3 | Do | W | + | | + | |
| Fer | 1060 | 10.0 | 19.5 | 6.8 | 0.33 | 10.1 | 0.21 | 43.0 | 0.10 | 47.2 | 18.1 | 8.1 | 304.0 | 102 | 7.4 | 15.6 | 36.9 | Do | W | + | | + | |
| Mit | 1082 | 3.3 | 4.7 | 4.7 | 0.10 | 7.5 | 0.13 | 34.0 | 1.23 | 38.5 | 18.4 | 8.2 | 345.8 | 101 | 18.4 | 22.6 | 17.4 | | W | | | | |
| Wild | 1136 | 2.3 | 2.4 | 2.4 | 0.22 | 8.6 | 0.19 | 53.6 | 0.35 | 31.6 | 19.5 | 8.6 | 296.8 | 77 | 20.9 | 21.2 | 31.4 | Do | W | | | | |
| Tau | 1138 | 3.6 | 14.6 | 4.3 | 0.66 | 4.4 | 0.20 | 126.9 | 0.49 | 40.4 | 4.1 | 8.0 | 246.8 | 64 | 5.0 | 21.0 | 177.4 | Ls | C | + | | | |
| Hoef | 1192 | 0.6 | 1.9 | 1.9 | 0.47 | 3.1 | 0.14 | 5.3 | 0.14 | 20.7 | 10.1 | 8.8 | 220.6 | 64 | 14.3 | 15.2 | 153.1 | | A | | | | |
| SieK | 1205 | 0.2 | 11.3 | 11.3 | 0.56 | 1.0 | 0.23 | 99.4 | 0.36 | 38.6 | 10.8 | 8.2 | 382.4 | 130 | 7.4 | 8.8 | 655.2 | | A | | | | |
| SieG | 1207 | 0.8 | 20.2 | 9.5 | 0.55 | 1.0 | 0.24 | 3.8 | 0.51 | 39.1 | 11.8 | 8.1 | 536.7 | 118 | 5.8 | 10.8 | 552.8 | | A | | | | |
| GruW | 1393 | 2.3 | 5.8 | 3.7 | 0.18 | 14.1 | 0.18 | 19.9 | 0.29 | 29.7 | 3.2 | 8.1 | 198.6 | 99 | 14.0 | 20.3 | 13.8 | Ma | V | | + | | |
| Roe | 1450 | 1.0 | 7.6 | 1.5 | 2.86 | 20.4 | 0.18 | 78.1 | 0.86 | 28.9 | 3.2 | 8.2 | 248.1 | 120 | 6.0 | 16.4 | 144.2 | Ls | V | | + | | |
| SoiS | 1458 | 4.0 | 8.2 | 3.2 | 0.61 | 7.9 | 0.27 | 13.4 | 0.33 | 29.8 | 3.9 | 8.3 | 222.1 | 112 | 7.3 | 15.0 | 78.7 | Ma/Ls | V | + | | | |
| GruO | 1474 | 3.5 | 6.9 | 5.0 | 0.20 | 8.3 | 0.13 | 60.8 | 0.10 | 26.5 | 1.4 | 8.5 | 194.5 | 112 | 14.1 | 17.5 | 31.3 | Ls | B | | | | |
| GaiU | 1508 | 3.5 | 4.1 | 3.8 | 0.25 | 12.0 | 0.29 | 9.6 | 0.18 | 16.4 | 10.1 | 8.9 | 198.3 | 72 | 13.4 | 16.3 | 21.3 | Do | A | | | | + |
| SoiN | 1520 | 0.3 | 4.7 | 4.2 | 0.77 | 10.8 | 0.24 | 13.9 | 0.37 | 36.3 | 6.8 | 8.2 | 251.8 | 115 | 5.4 | 13.4 | 72.5 | Mi | V | | | | + |
| SoE | 1552 | 3.0 | 5.5 | 4.5 | 0.61 | 3.9 | 0.18 | 49.3 | 0.10 | 31.1 | 6.6 | 8.8 | 177.7 | 123 | 7.5 | 15.0 | 168.2 | Ls | K | | | | |
| SoW | 1558 | 3.0 | 11.5 | 7.0 | 0.36 | 4.5 | 0.16 | 82.9 | 0.10 | 30.2 | 6.3 | 8.3 | 204.2 | 91 | 14.7 | 14.5 | 97.8 | Ls | K | + | | + | |
| DelN | 1600 | 0.6 | 1.3 | 1.3 | 0.37 | 5.7 | 0.38 | 15.1 | 0.39 | 35.1 | 2.8 | 8.2 | 223.3 | 81 | 11.9 | 12.6 | 67.7 | Ls | K | | | | |
| DelS | 1600 | 0.2 | 4.2 | 4.2 | 0.45 | 4.6 | 0.23 | 16.6 | 0.22 | 40.5 | 5.0 | 8.8 | 177.2 | 67 | 9.3 | 14.1 | 101.9 | Ls | K | | | | |
| Hoer | 1601 | 0.5 | 1.8 | 1.8 | 0.34 | 17.3 | 0.21 | 20.4 | 0.37 | 24.7 | 1.0 | 7.9 | 225.2 | 90 | 11.8 | 15.3 | 21.1 | Mi | A | | + | | |
| Fun | 1601 | 2.5 | 4.5 | 3.5 | 0.03 | 10.3 | 0.13 | 44.3 | 0.10 | 31.3 | 4.3 | 8.4 | 274.4 | 91 | 10.3 | 14.9 | 7.0 | Ls | B | | | | + |
| Seeb | 1657 | 6.3 | 18.4 | 6.9 | 0.52 | 4.5 | 0.14 | 23.3 | 0.14 | 27.0 | 4.4 | 8.8 | 165.9 | 82 | 6.8 | 13.8 | 121.6 | Ls | M | | | | |
| Scha | 1680 | 3.0 | 4.4 | 3.8 | 0.49 | 7.2 | 0.17 | 19.8 | 0.10 | 27.1 | 4.9 | 8.7 | 167.1 | 114 | 12.6 | 16.8 | 71.2 | Ls | W | | | | |
| Gug | 1725 | 0.1 | 1.9 | 1.9 | 0.29 | 4.6 | 0.23 | 10.6 | 0.86 | 15.0 | 10.3 | 9.0 | 207.3 | 100 | 11.8 | 13.2 | 64.7 | Do | A | | | | |

**Table 1.** *Cont.*

| Lake | Alt. m asl | Area ha | Depth m | Sec. m | NO$_3$ mg/L | TP µg/L | Si mg/L | NH$_4$ µg/L | Na mg/L | Ca mg/L | Mg mg/L | pH | Cond µS/cm | O$_2$ % | ABT °C | AST °C | DIN:TP | Geo. | Reg. | F | C | T | D |
|---|---|---|---|---|---|---|---|---|---|---|---|---|---|---|---|---|---|---|---|---|---|---|---|
| GaiO | 1769 | 0.8 | 2.9 | 2.7 | 0.15 | 3.7 | 0.24 | 25.3 | 0.18 | 14.6 | 7.0 | 8.5 | 201.0 | 114 | 9.5 | 16.3 | 46.8 | Do | A | | | | |
| Zie | 1799 | 3.0 | 15.1 | 5.0 | 0.14 | 7.6 | 0.15 | 26.7 | 0.10 | 22.5 | 1.6 | 8.2 | 219.8 | 107 | 6.7 | 13.7 | 21.5 | Ls | R | | | | |
| Seel | 1809 | 0.4 | 5.4 | 5.4 | 0.59 | 7.9 | 0.12 | 31.5 | 0.10 | 23.8 | 1.0 | 8.7 | 134.2 | 93 | 12.4 | 14.9 | 79.3 | Ls | B | | | | |
| Eis | 1827 | 0.7 | 3.9 | 3.9 | 0.24 | 1.1 | 0.21 | 23.3 | 0.36 | 16.0 | 3.6 | 8.2 | 192.7 | 96 | 6.4 | 10.4 | 247.6 | Ma | A | | | | |
| Dra | 1874 | 5.3 | 20.7 | 10.3 | 0.30 | 4.0 | 0.19 | 19.6 | 0.10 | 26.8 | 3.1 | 8.6 | 157.7 | 97 | 4.9 | 11.9 | 79.9 | Ls | M | | | | |
| Eng | 1876 | 3.0 | 17.3 | 10.4 | 0.04 | 4.7 | 0.59 | 27.0 | 0.31 | 19.8 | 7.2 | 8.2 | 235.6 | 89 | 4.9 | 11.1 | 14.4 | Do | A | + | | | |
| Bre | 1903 | 1.5 | 6.2 | 6.2 | 0.37 | 6.6 | 0.12 | 14.0 | 0.10 | 26.7 | 2.8 | 8.6 | 150.6 | 90 | 4.6 | 11.4 | 58.5 | Ls | M | | | | |
| Stu | 1921 | 3.0 | 5.1 | 5.1 | 0.25 | 7.7 | 0.12 | 19.6 | 0.10 | 23.3 | 1.6 | 8.8 | 126.0 | 112 | 12.4 | 16.7 | 35.4 | Ls | W | | | | |
| Lauf | 2012 | 0.8 | 5.6 | 3.7 | 0.13 | 3.8 | 0.22 | 24.9 | 0.19 | 12.4 | 6.3 | 8.3 | 168.2 | 99 | 9.4 | 14.7 | 39.1 | Do | A | | | | |
| Rap | 2047 | 2.3 | 7.8 | 5.0 | 0.04 | 9.2 | 0.17 | 15.9 | 0.23 | 21.2 | 9.3 | 8.6 | 205.1 | 101 | 11.0 | 15.6 | 6.5 | Do/Ma | A | | | | |
| Grub | 2060 | 0.5 | 3.5 | 3.2 | 0.10 | 17.0 | 0.15 | 17.8 | 0.18 | 37.9 | 2.6 | 8.6 | 222.4 | 73 | 9.4 | 13.3 | 6.9 | Ls | R | | | | |
| SeeU | 2224 | 2.4 | 1.7 | 1.4 | 0.07 | 2.5 | 0.25 | 21.4 | 0.99 | 19.8 | 9.1 | 8.8 | 208.6 | 113 | 16.1 | 17.0 | 35.7 | Do/Ma | L | | | | |
| Adl | 2294 | 1.9 | 1.7 | 1.7 | 0.15 | 1.0 | 0.13 | 30.3 | 0.32 | 19.4 | 9.2 | 8.5 | 159.6 | 94 | 14.0 | 14.7 | 183.3 | Do | L | | | | + |
| Schi | 2300 | 2.0 | 5.5 | 3.3 | 0.34 | 1.0 | 0.14 | 45.7 | 0.29 | 19.8 | 10.1 | 8.3 | 164.7 | 47 | 7.2 | 13.1 | 390.4 | Do | L | | | | |
| SeeM | 2424 | 0.5 | 4.1 | 4.1 | 0.20 | 1.5 | 0.13 | 67.8 | 0.14 | 17.6 | 9.9 | 8.7 | 137.8 | 122 | 10.6 | 11.5 | 173.2 | Do | L | | | | |
| SeeO | 2469 | 1.6 | 13.1 | 5.8 | 0.11 | 7.3 | 0.18 | 67.9 | 0.23 | 15.4 | 5.5 | 8.6 | 121.4 | 71 | 5.5 | 12.9 | 24.8 | Do | L | | | | |

## 2.2. Sampling and Laboratory Procedures

The 43 lakes were sampled twice during the ice-free period, once between June and mid-August and once between August and November. Hence, 36 lakes were investigated in 2016 and seven lakes in 2017. On the first sampling date, lake bathymetry was determined with an echo sounder (Lawrence HDS8, Oslo, Norway) and a buoy was subsequently installed, fixed to a stone at the deepest point of each lake by a rope. Temperature loggers were mounted on the rope, (Onset Pendant UA-001-64 HOBO, Bourne, MA, USA) 0.5 m above the ground and 0.5 m below the water surface. To assess the temperature regime and the mixing type of the lakes, the loggers were exposed in the lakes during most of the ice-free period. Physical parameters (temperature, oxygen saturation, pH, and electrical conductivity at 25 °C) were measured with a multiprobe (WTW 350, Weilheim, Germany) in one-meter steps above the deepest point of each lake on both sampling dates. After measuring the Secchi depth, 0.5 L of a mixed water sample was collected with a hose sampler from the euphotic zone [61]. One half of the water sample was filtered (0.45 μm) on-site and stored at 4 °C together with the unfiltered rest until further processing in the laboratory. Another liter of water was taken from the euphotic zone with the hose sampler and preserved with Lugol's solution to analyze planktic diatom communities [62]. Periphytic diatom assemblages were recorded by scraping the diatom communities off five stones, each taken at depths between 20 cm and 50 cm in the northern and southern littoral zone of each lake, with a single-use toothbrush [63]. Out of all of the sampled lakes, periphytic diatom samples could be obtained at 34 sites, while no stones were available at nine sites. On the second sampling date, sediment cores were taken from the deepest point of each lake with a gravity corer (Uwitec, Mondsee, Austria) to record the sedimentary diatom communities from all 43 lakes [28].

All chemical analyses were carried out in the laboratory of the Limnological Research Station Iffeldorf of the Technical University of Munich, Germany. Standard colorimetric methods were applied to determine the concentrations of total phosphorus [64], nitrate-N [65], ammonia-N [66], and silica (Nanocolor silica test, Macherey-Nagel, Düren, Germany). The concentrations of major ions (calcium, magnesium, and sodium) were measured using a cation chromatograph (Thermo Scientific, ICS-1100, Waltham, MA, USA).

Planktic diatom samples (1 L) were filtered with 0.45-μm syringe filters and the residue on the filters was further processed [67]. The uppermost centimeter of each sediment core was used to assess the sedimentary diatom assemblages. The residue on the filters of the planktic samples, the sediment samples, and the littoral samples were processed in the same way: The diatoms were prepared according to van der Werff and Macan [68]. To analyze the composition of the diatom samples, 500 valves were identified in each case using a Leica DNM microscope (Wetzlar, Germany) at 1000× magnification. Eleven of the 43 planktic samples were excluded from the data analysis because they contained an insufficient number of valves. Taxa were counted at the species level and, if possible, at the subspecies level. Individuals that could not be identified were given working names. Standard literature was used for identification [69–76].

## 2.3. Data Analysis

Since periphytic diatom assemblages can vary significantly within a lake [77,78], littoral samples from the northern and southern shore of each lake were pooled to obtain a more representative indication of the periphytic diatom assemblage. All statistical analyses were computed for the sedimentary data set (N = 43), the littoral data set (N = 34), and the planktic data set (N = 32) analogously.

The following parameters were computed to capture the conservation value of each sample using the current German Red List for diatoms [41]. Community indices were calculated based on the taxa with a status higher than "not in danger/insufficient data" ($RL_D$), overall species richness per sample (N), overall abundance per sample (Ab), and overall abundance of all $RL_D$ ($Ab_{RL}$):

N_rl = sum of $RL_D$
share_rl = N_rl/N
rel_share_rl = $Ab_{RL}$/Ab
weight_rl = rel_share_rl weighted by Red List class

Diatoms were assigned weighting factors ranging from 1 to 7, according to the threat category ranging from "warning list" to "threatened by extinction", to calculate the weight_rl. Environmental predictor variables were selected based on their variance inflation factors (VIF). These were calculated with the "vif" function from the R package "usdm" [79]. To reduce multicollinearity among environmental predictors, only variables with VIFs less than five were included for each individual dataset. Shannon diversity was calculated within the "vegan" [80] package in R. The uniqueness of diatom assemblages was calculated as the local contribution to beta diversity (LCBD) at each site for Hellinger-transformed abundance data using the R package "beta.part" [45]. This approach is neutral with regard to ecological prerequisites (e.g., $\alpha$ and $\gamma$ diversity) and allows the mathematically correct identification of assemblages' uniqueness within a larger metacommunity. The total variance of a community matrix (Var(Y)) equals the overall $\beta$ diversity ($BD_{Total}$). This value is the total sum of squares ($SS_{Total}$; the sum of squared deviation for all sites and species from the mean) divided by $n_{Sites}$—1. The LCBD value for each sampling unit is derived by dividing the sum of squares of each site (SSi) by the total sum of squares ($SS_{Total}$). Accordingly, large LCBD values indicate a high contribution of a site to the overall $\beta$ diversity. LCBD was partitioned in the component reflecting species substitution (Replacement; "Repl") and the component reflecting species loss (Nestedness; "Nest") [81]. The replacement component was the main contribution to $BD_{total}$ for all datasets (sediment: 99.5%, $BD_{total\_sed}$ = 0.385; littoral: 99.6% $BD_{total\_lit}$ = 0.282; plankton: 97.0%, $BD_{total\_pla}$ = 0.359).

In order to test and visually verify the strength of the correlation between RL indices and the selected environmental and community-based predictor variables, a correlation plot was generated using the "corrplot" package in R [82]. To account for multiple testing, *p*-values were Bonferroni-corrected a priori. Previous work revealed pure spatial effects and shared effects of environmental and spatial variables on diatom communities within the lake set [83]. Therefore, the spatial influence on all response and predictor variables was assessed by calculating Moran's I autocorrelation coefficient based on latitudes and longitudes recorded for each lake using the R package "ape" [84].

The normal distribution of response variables was tested by the Shapiro–Wilk test in the "vegan" package. Linear models (hereafter called "LMs") were applied for normally distributed variables that were spatially structured according to Moran's I *p*-values. The spatial structure of model residuals was tested with the "lm.morantest" function from the "spdep" package [85]. No spatial structure of the residuals was detected in any of the computed LMs. If variables had a non-normal distribution and no spatial autocorrelation was indicated, generalized linear models (hereafter called "GLMs") were applied within the "vegan" package. Diagnostic plots were checked for structures within model residuals. Generalized additive models (hereafter called "GAMs") were developed, and spatial predictors were included (longitude and latitude) within the package "mgcv" [86] in cases of non-normal distributed and spatially autocorrelated variables. Variables were fitted with smoothers and k values and regression splines were fitted to increase model suitability. Non-significant variables were excluded until each predictor was significant. Reduced and original models were compared based on Akaike information criterion (AIC) [87] and the most parsimonious model was selected.

## 3. Results

Briefly, 560 diatom taxa were recorded across the three diatom assemblages of the 43-lake dataset and 360 taxa (64%) were on the German diatom RL. Of these, 184 (51%) were assigned a threat status ranging from "warning list" to "threatened by extinction". Across the subsets, 276, 240, and 139 taxa were on the RL for sedimentary, littoral, and

planktic assemblages, respectively. Of these 135, 122 and 64 species were assigned a threat status for sedimentary, littoral, and planktic assemblages, respectively (Figure S2).

Significant and ecologically meaningful correlations between RL indices and predictor variables were revealed by Pearson correlations (Figure 2). TP was significantly and negatively correlated with rel_share_rl for all assemblages. Both variables were spatially structured (Moran's $p < 0.05$, Figure 3a), but no such structure was found in the LM residuals ($p = 0.175$, Table 2), suggesting that the spatial structure of both variables was congruent. LMs confirmed the correlation between TP and rel_share_rl for sedimentary diatoms ($p = 0.002$, $R^2 = 0.20$, Figure 4a, Table 2).

For littoral diatoms, the correlation of TP with rel_share_rl became non-significant in the LM when Secchi depth was included ($p_{TP} = 0.182$; $p_{secchi} = 0.029$, Table 2), probably reflecting the intercorrelation of both variables as predictors of the lake's trophic state. After excluding TP, the LM confirmed the positive correlation of rel_share_rl and Secchi depth for the littoral assemblages ($p = 0.009$, $R^2adj = 0.17$, Figure 4b, Table 2). Both variables were spatially structured (Moran's $p < 0.05$, Figure 3b) but model residuals were not, indicating congruence of their distribution in space ($p = 0.607$, Table 2). TP was not correlated with any other variable in the dataset and Secchi depth was only correlated with August bottom temperature, which was not significantly correlated with any other variable (Figure 2). The significant correlation of TP and rel_share_rl within planktic assemblages turned out to be mediated by the longitude of the sampled lakes ($p_{lon} < 0.001$, $p_{lat} = 0.117$, $p_{TP} = 0.327$), and the final model incorporated longitude as the sole predictor ($p < 0.001$, $R^2 = 0.34$) (Table 2, Figure 4c). Accordingly, a marked increase of rel_share_rl could be observed in the "Allgäu" and "Lechtaler Alpen" lakes in the western part of the study region (Figure 4c, left of the vertical dashed line; see Table 1 for background information). Finally, magnesium content was positively correlated with RL indices for all datasets, and it was intercorrelated with conductivity in the sedimentary dataset and with TP in the planktic dataset ($p < 0.05$) (Figures 2 and S1).

**Table 2.** A significant correlation was detected based on linear regression models (LMs) between the share of RL species (share_rl) and total phosphorous (TP) and Secchi depth (secchi) for sedimentary and littoral assemblages, respectively. For planktic assemblages, share_rl significantly decreased with longitude (lon) based on GAMs. *p*-values for the Moran-test of LMs residuals and AIC values are given where relevant. Within GAMs, inclusion of smoothers is indicated by placing the predictor within "s()".

| Response Share_rl | $p_{Shapiro}$ | Model Type | Predictor | $p_{model}$ | $R^2adj$ | Res. Df | $p_{lm.moran}$ | AIC |
|---|---|---|---|---|---|---|---|---|
| Sediment | 0.098 | LM | TP | 0.002 | 0.20 | 41 | 0.175 | |
| Littoral | 0.931 | LM | TP | 0.056 | 0.08 | 32 | 0.908 | |
| | | LM | TP + secchi | TP: 0.182 secchi: 0.029 | 0.19 | 31 | 0.923 | |
| | | LM | secchi | 0.009 | 0.17 | 32 | 0.607 | |
| Plankton | 0.02 | GAM | s(lon) + s(lat) + s(TP) | lon: <0.001 lat: 0.117 TP: 0.327 | 0.38 | 19 | | −38.4 |
| | | GAM | s(lon) | <0.001 | 0.34 | 27 | | −41.3 |

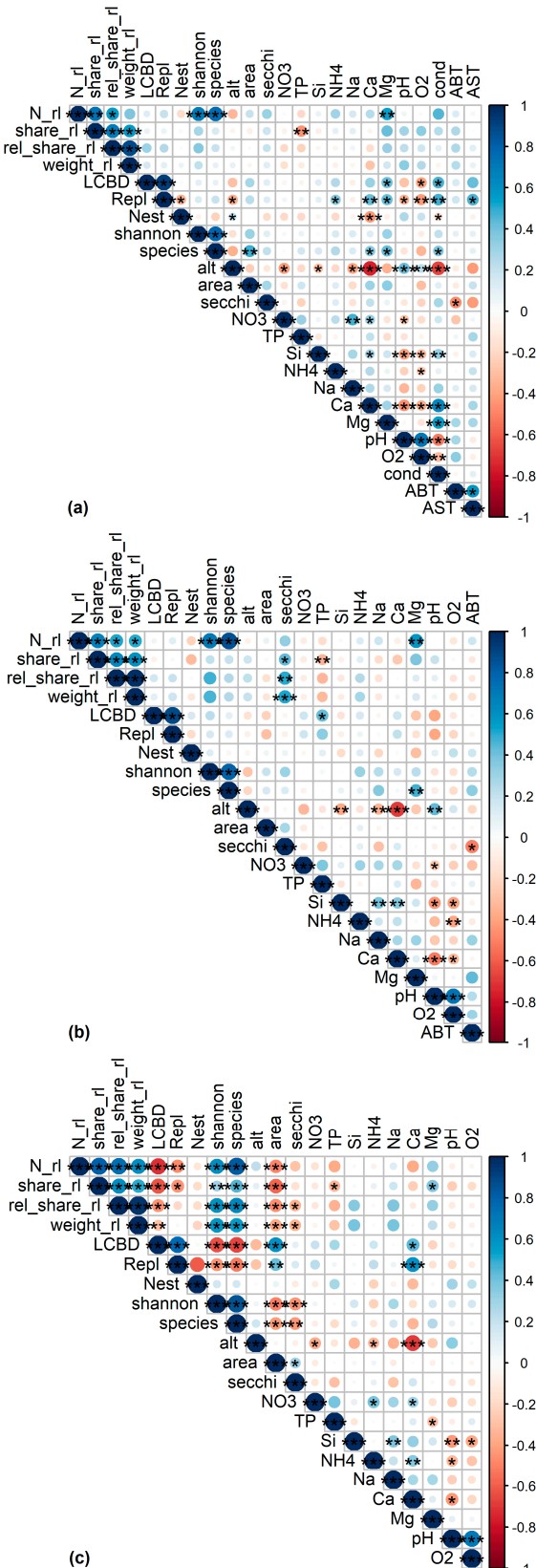

**Figure 2.** Correlations plots for the (**a**) sedimentary (N = 43), (**b**) littoral (N = 34) and (**c**) planktic (N = 32) diatom dataset. Abbreviations of environmental parameters are explained in Table S1. Asterisks: *** $\triangleq$ *p* value < 0.001; ** $\triangleq$ *p* value < 0.01; * $\triangleq$ *p* value < 0.05. Point color refers to the Pearson correlation coefficient.

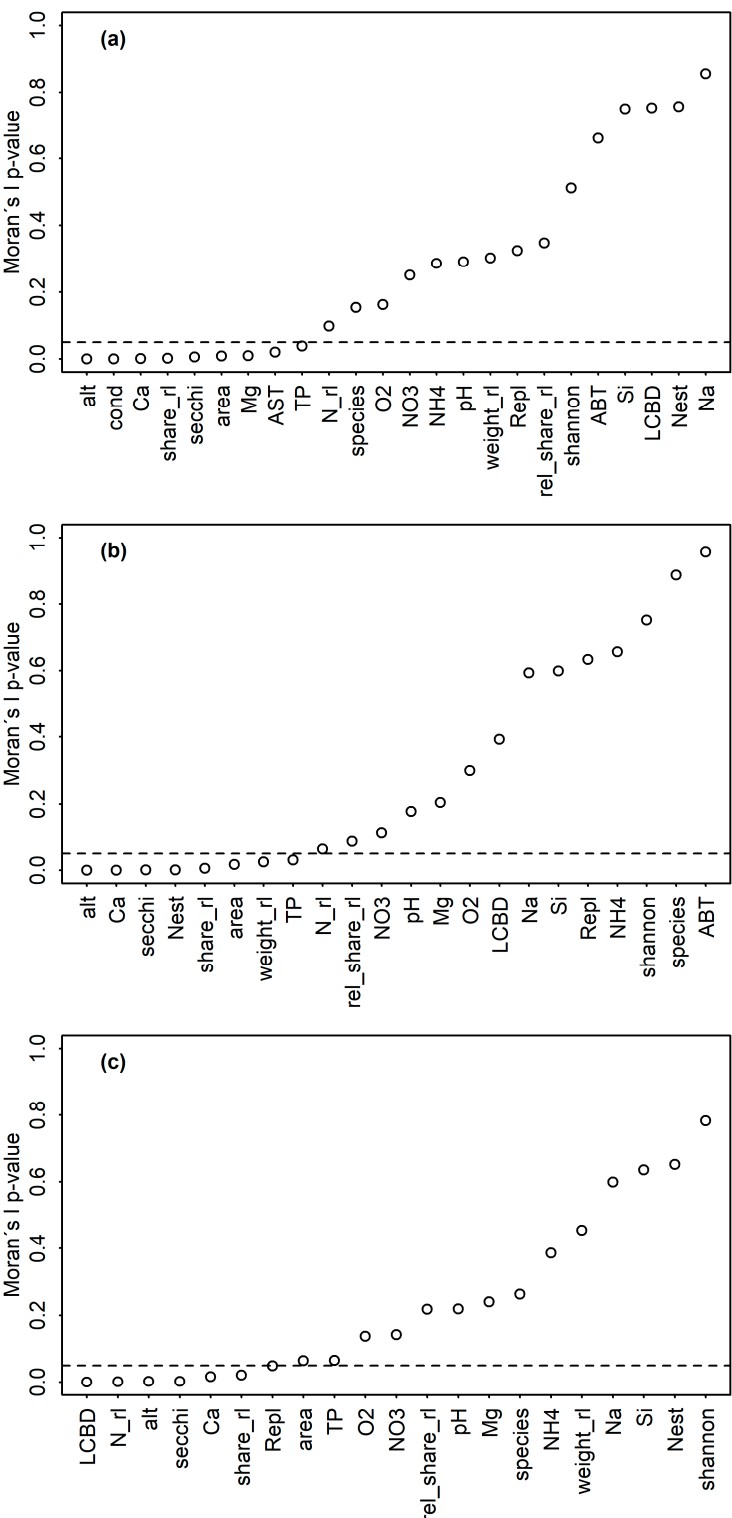

**Figure 3.** Moran's I *p*-values for RL indices, assemblage indices and environmental variables plotted in increasing order for the (**a**) sedimentary (N = 43), (**b**) littoral (N = 34) and (**c**) planktic (N = 32) dataset. The horizontal dashed line marks a *p*-value of 0.05, with lower values indicating spatial autocorrelation based on longitudes and latitudes. Abbreviations of environmental parameters are explained in Table S1.

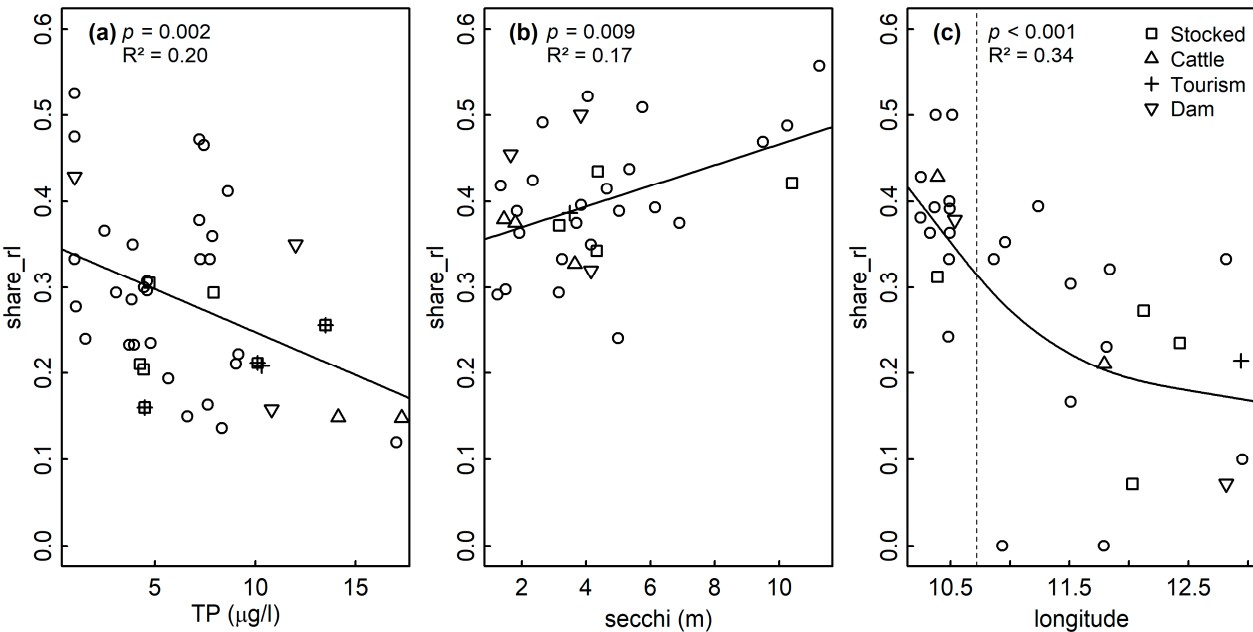

**Figure 4.** A significant correlation was detected based on LMs between the share of RL species (share_rl) and total phosphorous (TP) and Secchi depth (secchi) for (**a**) sedimentary and (**b**) littoral assemblages, respectively. A significant decrease of share_rl with longitude (lon) was found for planktic assemblages (**c**) based on GAMs.

GAMs and LMs confirmed the significant and positive correlation between species richness and N_rl, as indicated by Pearson correlations for sedimentary diatoms ($p < 0.001$, $R^2$adj = 0.56) and littoral diatoms ($p < 0.001$, $R^2$adj = 0.74), respectively (Table 3, Figure 5a,b). Both variables were not spatially structured within the sedimentary dataset (Moran's I $p > 0.05$). No spatial structure was observed in the model residuals ($p = 0.219$) for the littoral dataset, and GAMs revealed an increase of RL species richness (N_rl) for planktic diatoms, with overall species richness and a negative correlation of N_rl with longitude ($p < 0.001$, $R^2 = 0.81$) (Table 3, Figure 5c).

**Table 3.** A significant correlation between the number of RL species (N_rl) and species richness was detected for all assemblages based on LMs, GLMs, and GAMs. For planktic assemblages, longitude (lon) was additionally correlated to N_rl. *p*-values for the Moran-test of the linear model residuals and AIC values are given where relevant. If smoothers were included in GAMs, this is indicated by placing the predictor within "s()".

| Response N_rl | $p_{Shapiro}$ | Model Type | Predictor | $p_{model}$ | $R^2$adj | Res. Df | $p_{lm.moran}$ | AIC |
|---|---|---|---|---|---|---|---|---|
| Sediment | <0.001 | GLM | species | <0.001 | 0.56 | 41 | | |
| Littoral | 0.083 | LM | species | <0.001 | 0.74 | 32 | 0.219 | |
| Plankton | 0.092 | GAM | s(lon) + s(lat) + s(species) | lon: 0.008 lat: 0.826 species: <0.001 | 0.80 | 28 | | −63.00 |
| | | GAM | s(lon) + s(species) | lon: <0.001 species: <0.001 | 0.81 | 29 | | −63.02 |

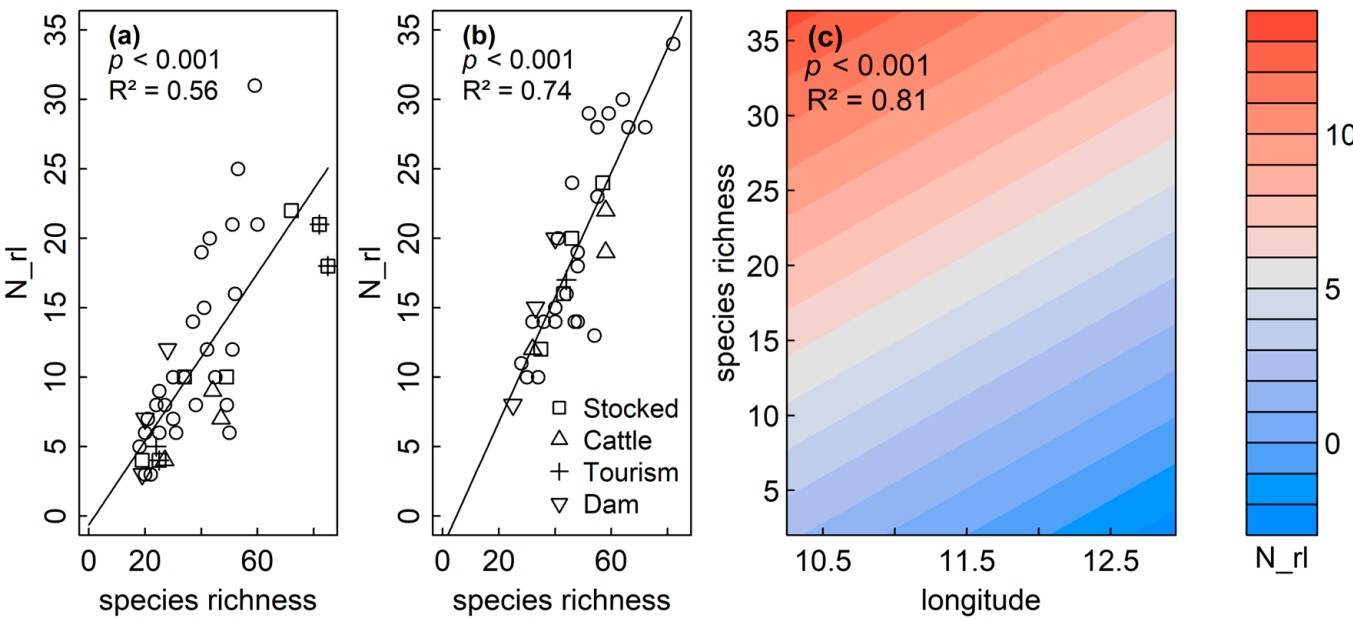

**Figure 5.** The number of Red List species (N_rl) increased with species richness for sedimentary (**a**), littoral (**b**) and planktic (**c**) assemblages. Longitude (lon) was additionally negatively correlated to N_rl for planktic assemblages. The color indicates the number of RL taxa.

A positive correlation of assemblage uniqueness (LCBD) and rel_share_rl was revealed by GAMs for sedimentary diatoms ($p = 0.049$, $R^2 = 0.07$). Omitting one outlier lake with extremely high conductivities due to groundwater influence (SieG, Table 1), high LCBD values resulted in an increasing and a decreasing branch of rel_share_rl values along the LCBD gradient (Figure 6a). A common feature of five lakes of the lower branch (Bic SoW, Tau, Laut, Fer) was that they are impacted by fish introduction and in three cases (SoW, Laut and Fer) by huts, hotels, or restaurants on the shoreline. The negative relationship between LCBD and rel_share_rl for planktic assemblages ($p = 0.010$, $R^2 = 0.18$, Table 4, Figure 6c) probably reflects the intercorrelation of LCBD with species richness ($p < 0.001$), which in turn was highly correlated with rel_share_rl ($p < 0.001$, Figure 2). No significant correlation of uniqueness and RL indices was found for the littoral diatoms ($p > 0.05$, Table 4, Figure 6b).

**Table 4.** A significant correlation between the abundance share of RL species (rel_share_rl) and the local contribution to β diversity (LCBD) was revealed by GAMs for sedimentary and planktic assemblages.

| Response Rel_Share_rl | Model Type | Predictor | $p_{model}$ | $R^2adj$ | Res. Df |
|---|---|---|---|---|---|
| Sediment | GAM | LCBD | 0.049 | 0.07 | 41 |
| Littoral | GAM | LCBD | 0.420 | 0.02 | 32 |
| Plankton | GAM | LCBD | 0.010 | 0.18 | 30 |

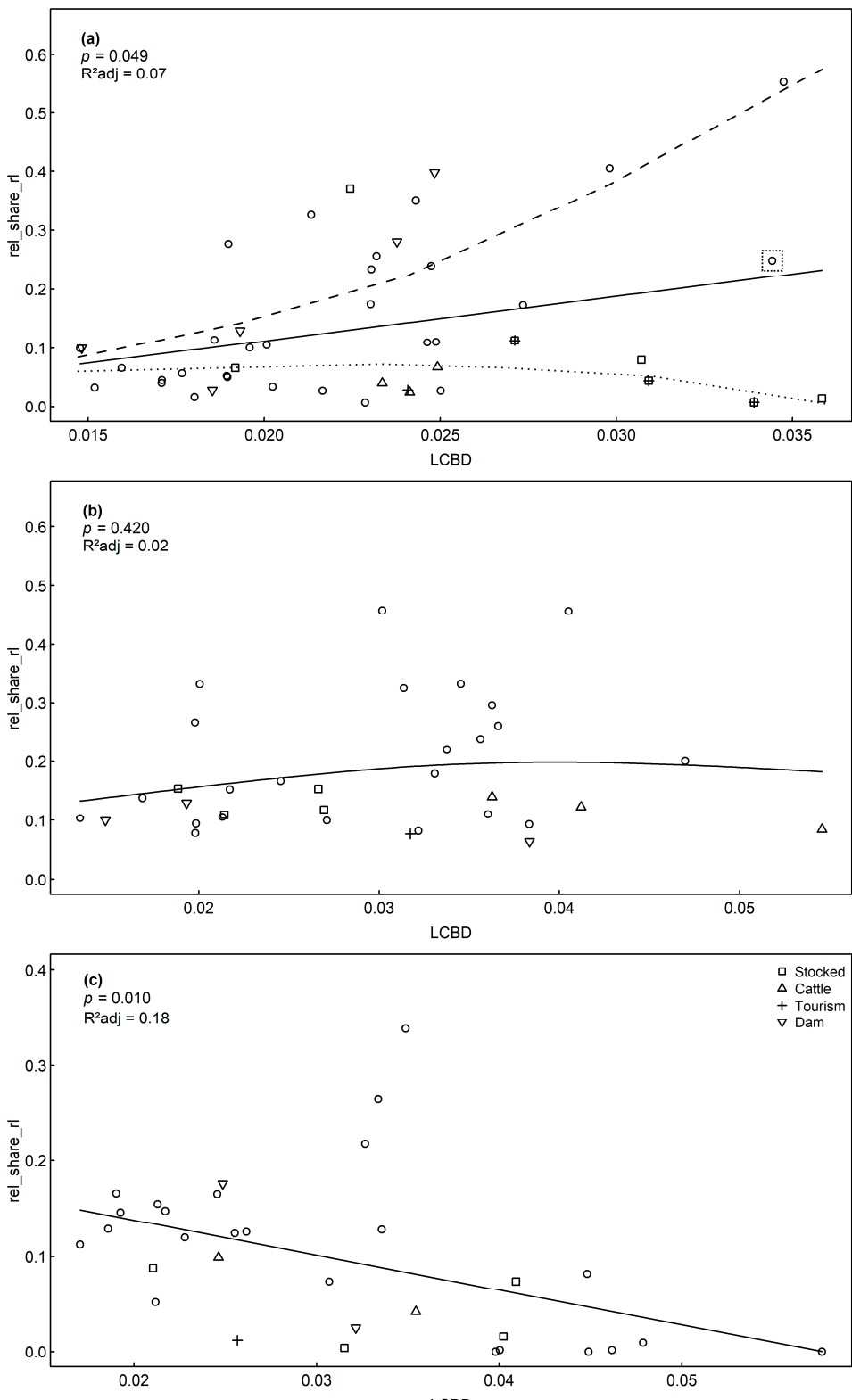

**Figure 6.** The abundance share of RL species (rel_share_rl) increases with higher uniqueness (represented by the local contribution to β diversity, LCBD) for sedimentary (**a**) and decreases for planktic assemblages (**c**), while no significant trend was found for littoral assemblages (**b**). The assumed trend for unimpaired and impaired lakes is indicated by a dashed and dotted line respectively for the sedimentary dataset. The dotted box indicates one outlier lake with a strong influence of groundwater (SieG, Table 1).

## 4. Discussion

This study revealed a high proportion and abundance of endangered diatoms in naturally oligotrophic, fishless mountain lakes and underlined their vulnerability to eutrophication. This finding could be observed for sedimentary, littoral and planktic assemblages, supporting the broad applicability of the German Red List (RL) for diatoms [41] and confirming previous research on the effects of eutrophication on mountain waters [88]. A new approach was taken by comparing the uniqueness of assemblages and diatom RL indices. Unique sedimentary assemblages revealed either pristine lakes with high abundances of RL taxa or lakes with fish stocking and low RL species abundance. In turn, α diversity reflected only RL richness, but not the share or abundance of RL taxa, thereby losing important information on mountain lake conservation value.

### 4.1. Nutrients and the Share of Red List Diatoms

Previous research associated high numbers of RL diatom taxa with oligotrophic or dystrophic freshwater habitats [49], reflecting the general rarity of these environments in central and southern Europe [48]. In mountain regions such as the European Alps, oligotrophic habitats are still abundant, but they are often threatened by eutrophication. This is especially true of springs [89] and lakes [53]. Our results substantiate these findings by revealing a negative correlation of the share of RL taxa with TP for sedimentary assemblages. Moreover, a positive correlation between Secchi depth and RL share was found for littoral assemblages. Both findings indicate the suppression of oligotraphentic diatom taxa at higher trophic levels. In the lakes of our dataset, the elevated TP levels of three subalpine lakes (Roe, Hoer, GruW; Table 1), which were surrounded by pastures, are likely caused by intensive cattle herding. Strong trampling of the grass layer and excrements near the lakes probably led to the observed eutrophication [19]. Another important nutrient source includes mountain huts and touristic infrastructure, such as hotels and restaurants [20,90,91], associated with four lakes in our dataset (Laut, Fer, Fun, SoW; Table 1). Furthermore, damming can lead to soil mineralization, possibly leading to the eutrophication of two subalpine lakes (GaiU, SoiN) in our dataset, while a dammed, high-altitude lake (Adl) remained nutrient poor. This may be related to the thin soil layer in the alpine region [92] and the consequently lower mineralization potential. In contrast to previous research on RL diatoms in springs [49], the negative correlation of trophic level and share of RL taxa was attributed to TP rather than nitrate. This difference may be due to phosphate limitation of the study lakes, indicated by DIN/TP ratios above 3.4 [93] (Table 1, range: 7–655). If nitrite had been measured as a further component of DIN in addition to nitrate and ammonium, the N/P ratios would have been even higher. High concentrations of inorganic nitrogen in mountain lakes are related to atmospheric dry and wet deposition [94–96]. This atmospheric input may be less pronounced in springs due to their smaller surface. Springs may therefore be nitrogen co-limited, causing stronger responses to a shift in nitrate concentrations. Moreover, some of the studies that identified nitrate as an important correlate of RL indices did not include TP measurements [97], which may be autocorrelated with nitrate. Nevertheless, there is empirical evidence that elevated nitrate levels caused by direct loadings are not always concomitant with an increase of TP in springs [55]. Overall, our findings confirm the hypothesized negative correlation between the RL share and the trophic state, while the number of RL taxa and abundance were not correlated. Our conclusions are in line with those of previous work, namely that mountain waters need to be better protected from human-induced nutrient inputs to conserve rare diatom taxa and thereby sustain valuable habitats for a broader range of biota. An unexpected observation was the positive correlation of magnesium (Mg) with RL richness across all assemblages (Figure S1). Mg is known to influence diatom community composition, e.g., in springs from the Apennines [55] or in petrifying springs from Lower Belgium [98]. Moreover, some diatoms are closely bound to alkaline conditions with high Mg contents, such as *Achnanthidium dolomiticum* Cantonati & Lange-Bertalot [54,76]. Generally, we found high Mg contents in lakes with dolomite-dominated catchments, corre-

sponding to the chemical composition of dolomite ($CaMg(CO_3)_2$). This probably explains the higher number of RL taxa in the western region, which is rich in lakes on dolomite bedrock. Previous studies have found a correlation between geodiversity and biodiversity of springs [55] and streams [99]. In our dataset, RL richness was positively correlated with total diatom richness, suggesting a possible indirect effect of geodiversity on RL diatom richness. These results indicate the need for further research on the potentially important role of dolomite-dominated catchments for diatom α diversity.

*4.2. Uniqueness as an Indicator of Fish Stocking*

The correlation of RL species abundance and uniqueness was relatively weak for sedimentary assemblages ($R^2 = 0.07$). This does not necessarily indicate that the metric is of no use for detecting rare and restricted taxa, but reflects the potential of LCBD to indicate both pristine and disturbed sites: For fish in the Doubs river in France, the highly unique sites were either undisturbed and environmentally unique and corresponded to headwaters with steep slopes or impaired by eutrophication [45]. As hypothesized, a similar pattern was found for the sedimentary diatom assemblages in our dataset. Whereas the abundance share of RL taxa generally increased from low to moderate LCBD values, the most unique sites showed pronounced differences with respect to RL abundances: a group of lakes that is stocked with fish was highly unique, but had low RL abundances (<10%), while no direct human pressures are known for two other highly unique lakes, reflected by very high RL abundances (>40%). This picture is exemplified by the extreme difference in RL abundance between the adjacent lakes SoE (rel_share_rl: 40.5%) and SoW (rel_share_rl: 0.7%), which are only about 100 m apart. During hut construction in 1866 on the shore of SoW, the lake was stocked with *Salvelinus alpinus* Linné. SoE was not stocked as it experiences strong changes in water level of up to 9.5 m [20], probably contributing to its ecological distinctness [100,101]. This suggests that ecological uniqueness is indicative within sedimentary diatom assemblages for degraded or pristine and environmentally unique sites. Importantly, not all pristine sites were unique, reflecting the context dependency of the metric [47]: unaffected sites with low TP were frequent in our data set. Therefore, only those lakes that were additionally distinct, e.g., due to water level fluctuations, had unique assemblages. This is also reflected by one outlier lake (SieG) that has been strongly influenced by groundwater and is very deep relative to its surface, probably leading to lower RL abundances in the sedimentary assemblage.

Our results confirm the hypothesized potential of LCBD to identify assemblages from pristine lake that are environmentally distinct and those from degraded lakes. Thus, a sound interpretation of the LCBD index is only possible when backed up with basic environmental data and information on human influence. If this is the case, it can be helpful in detecting complex ecological interactions, such as the effects of fish stocking in mountain lakes.

Degradation related to fish stocking may be due to either a change of the nutrient cycle, i.e., bottom-up effects [23], top-down control due to the selective impact on planktic and benthic invertebrates [102–105], or through a combination of both [106]. Of the seven fish-stocked lakes in our dataset, only those lakes that are additionally impacted by intensive tourism ("Lau", "Fer") were TP enriched. This indicates that predation is mainly on zooplankton rather than zoobenthos for the stocked lakes, which would cause the additional introduction of benthic nutrients into the open water [23], manifested in elevated TP levels. Permanent fish stocking can contribute to the eutrophication of lakes through the addition of biomass that will be recycled after the fish die [107]. Moreover, Pastorino et al. [108] found more than 80% of terrestrial invertebrates in the stomach of introduced *Salvelinus fontinales* Mitchill from an alpine mountain lake, possibly leading to additional allochthonous biomass input. However, these processes should also be reflected in the TP level, and thus appear of minor importance in the study lakes that are mainly at lower elevations. Nonetheless, fish feeding on zooplankton also leads to a transfer of nutrients to algae from previously inaccessible sources, i.e., nutrients stored within the zoo-

plankton [109,110]. As a consequence, primary production will increase [23]. The observed low RL abundances in the fish-stocked lakes may be due to this eutrophication effect, which cannot be detected by TP measurements as the total phosphorous content in the pelagic zone remains unchanged. The stocked lakes in our dataset commonly contained moderate to high abundances of the benthic diatoms *Staurosirella pinnata* (Ehrenb.) D.M.Williams and Round and *Staurosira venter* (Ehrenberg) Cleve & Moeller that are tolerant to low light intensity (range of summed relative abundance: 4–53%) [111]. This indicates an enhanced pelagic primary production leading to shading of the benthic environment. To enhance the assessment of primary production, further research in stocked lakes should include measurements of algal pigments such as chlorophyl a, which were found to be a strong predictor of fish stocking effects on primary production [23].

The reason for the unique sedimentary diatom assemblages may be top-down control, i.e., the predation of planktivorous fish on zooplankton and consequently altered feeding patterns on phytoplankton [112]. In oligotrophic systems, top-down control often suppresses large zooplankton such as Daphnids [113,114] while it may promote predation-resistant small zooplankton such as rotifers and cyclopid copepodes [53,113]. The abundance of the zooplankton in turn depends on the trophic state, and a unimodal relationship along the TP gradient was found for Daphnids [103]. Thus, cascading effects on the phytoplankton are complex and likely to be dependent on the trophic level. While most studies find an effect on phytoplankton species traits and assemblage composition [114,115] due to the well-documented reduction in size of the zooplankton [104,114,116], the response of phytoplankton biomass varies and is dependent on the trophic state of the lakes [103,106,114,117]. The effect on species composition is indicated through the presence of the planktic diatom *Asterionella formosa* Hassall in three of the stocked lakes (relative abundance: 3–16%). This taxon is not edible by small zooplankton due to its large size and formation of colonies [118,119]. In turn, the most characteristic feature was the high planktic diatom proportion of small centrics (four lakes with > 50% centrics), a pattern documented after the extirpation of large zooplankton [105]. Within our lake set, this may be caused by enhanced nutrient recycling, which in turn may promote small centrics such as *Cyclotella comensis* Grunow, coupled with a stable water stratification [120]. The high share of centrics and *A. formosa* Hassall probably explains the high uniqueness of the stocked lakes as most other lakes had lower abundances of centrics. The only stocked lake with low LCBD values had a low share of centrics (Eng, 3% of centrics). Within this alpine lake, low water temperatures and low lake productivity may lead to low fish-density, enabling the coexistence of large zooplankton with fish [121]. Moreover, weaker top-down control may be due to the naturally low density of zooplankton [106] in ultraoligotrophic lakes, possibly leading to the previously observed feeding of fish on terrestrial insects that settle on the water surface [103,106,108,114,117]. It remains to be seen whether this possible eutrophication effect will turn the tide for primary producers in this lake, especially since secondary effects through more stable lake stratification at higher water temperatures may come into play [120]. To conclude, the response of the sedimentary diatom assemblages to fish stocking in our dataset may be triggered by both altered nutrient cycling and top-down effects. The change of nutrient cycling may lead to bottom-up effects that are hypothesized to interact with the top-down control of fish [106,122].

Unexpectedly, the top-down control in stocked lakes was not reflected by a high LCBD and low RL abundance of planktic samples. This may be due to the more volatile character of the lake plankton: short-term interference to the planktic assemblages through the introduction of benthic diatoms is frequent and depends on the mixing state [123,124]. The high rates of benthics in some samples prove a high importance of this process in our dataset. Moreover, the strength of fish-mediated pelagic nutrient recycling also varies during the growing season, with the highest rates occurring during summer, i.e., along with the highest water temperature [23]. Our sampling period lasted from June until November, inherently producing temporal variability within the planktic and littoral datasets. Another reason may be that only five true planktic species had a threat status, making inferences

based on planktic assemblages more difficult. Thus, the uniqueness of sedimentary diatom samples can be used more reliably to detect effects of fish stocking due to their time integration of environmental conditions [83,125]. Therefore, littoral diatom samples did not track fish-induced processes in our dataset either. Apart from the temporal variability, this may be due to stronger predation of fish on zooplankton than on zoobenthos. As the predation of zoobenthos increases with fish density [23], the response of littoral diatoms will be more pronounced in eutrophic and low altitude lakes, especially if stocked with bottom-orientated fish.

Overall, the results suggest that introduced fish have a severe impact on mountain lakes in the study region, justifying a halt to further stocking and the eradication of introduced fish, e.g., using gill nets [126]. The success of such measures has been seen in the Italian Alps and northern America [126,127], where the natural trophic structure reappeared only a few years after lakes became fishless, followed by a recolonization with the original species [128]. Finally, the eradication of fish in naturally fishless lakes will also help to restore associated assemblages outside the aquatic trophic pyramid, such as herpetofauna, which has been impaired or even eradicated in stocked lakes [129–131], and is one of the most threatened groups of animals [132,133].

*4.3. Diatom Richness and Conservation Value*

Diatom richness was positively correlated with RL richness in all assemblages, and it was additionally negatively correlated with longitude for planktic assemblages. Differences were detected in the impact of the fish-stocked lakes on the respective correlations. For littoral diatoms, the variance of the regression was low, and the stocked lakes were close to the linear regression line. On the other hand, RL richness within sedimentary diatoms of stocked lakes was lower than would be expected by the linear regression model. This suggests that in the regression of RL richness on species richness, the model coefficient, i.e., the slope of the regression line, and the structure within the model residuals are more informative for sedimentary than for littoral assemblages. This may indicate the conservation value of a lake. However, the hypothesized logarithmic correlation was not detected for any of the assemblages, probably because TP was not correlated with total species richness in our dataset. Therefore, the correlation of RL richness and species richness appears to be a rather weak metric of eutrophication of mountain lakes. As for habitat quality, the share of RL taxa was the only metric to track lake eutrophication. Thus, species richness is also unable to exploit the full potential of the diatom Red List, namely detecting rare habitats [48], as it was not correlated to the share of RL taxa. Preserving a high number of RL species is sometimes used as a stand-alone conservation objective and species richness can be used to assess high numbers of RL species in our dataset. However, this is probably not the most efficient way to sustain assemblages of rare taxa in the long run, as overall species richness was not significantly correlated with the abundance of RL species in sedimentary and littoral assemblages. A species is probably more likely to become extinct with a smaller population size, due to the pronounced effects of ecological drift [134,135]. Instead, the combination of species richness and uniqueness is suggested, both of which are complementary components of the assemblages RL status, i.e., RL richness and RL abundance.

**5. Conclusions**

This study substantiates the high conservation value of mountain lakes as documented by a high number, share, and abundance of endangered diatom taxa. The ecological vulnerability of the studied lakes in light of eutrophication and fish stocking was demonstrated through decreased shares and abundances of Red List taxa respectively. It is likely that entire lake ecosystems may be altered through the effects of eutrophication, due to external sources and top-down control as well as a change of nutrient cycling through introduced fish. This necessitates restoration measures, such as a stop to fish stocking, lake fencing, a decrease of cattle densities, and mitigation of infrastructure impacts. The results call for the

better inclusion of small mountain lakes within legal frameworks such as the European Water Framework Directive. Moreover, our findings suggest the broad applicability of diatoms as bioindicators. Their power to identify endangered freshwater habitats and threats to their biodiversity significantly enhanced by consideration of regional Red Lists of diatoms. It is important to calculate the share of Red List taxa and their abundance per sample as these parameters provide important information on impairments and the conservation value of lakes, compared to the sheer richness of Red List taxa. From a methodical point of view, diatom-based biomonitoring should include sedimentary samples wherever feasible, as the temporal integration of environmental conditions makes them the most robust indicators of eutrophication or fish stocking. Further research may reveal whether multiple sampling of the littoral and pelagic zone leads to converging results between both methods. Finally, diatom α diversity and assemblage uniqueness, i.e., the local contribution to β diversity, were successfully applied to assess the calculated Red List indices. Both components of species diversity contribute complementary information about the conservation value of aquatic ecosystems. At least basic information regarding human disturbance and environmental conditions is necessary for a sound interpretation of assemblage uniqueness.

**Supplementary Materials:** The following supporting information can be downloaded at: https://www.mdpi.com/article/10.3390/d14050389/s1, Table S1: Average values and variation of the assessed environmental parameters; Figure S1: The number of RL species (N_rl) increases along the magnesium (Mg) gradient for sedimentary (a) and (b) littoral assemblages; no significant relationship was found for planktic assemblages (c). Model parameters are given for GAMs; Figure S2: The number of diatom taxa according to their threat status within the German Red List (2018) is given for (a) the complete data set, (b) sedimentary assemblages, (c) littoral assemblages, (d) planktic assemblages and (e) true planktic assemblages.

**Author Contributions:** Conceptualization, S.O., J.G. and U.R.; data curation, S.O.; investigation, S.O. and A.M.H.; methodology, S.O.; supervision, J.G. and U.R.; writing—original draft, S.O.; writing—review & editing, S.O., A.M.H., J.G. and U.R. All authors have read and agreed to the published version of the manuscript.

**Funding:** S.O. was supported through a scholarship by the Deutsche Bundesstiftung Umwelt for the first author (Grant 20016/434).

**Institutional Review Board Statement:** Not applicable.

**Informed Consent Statement:** Not applicable.

**Data Availability Statement:** The datasets generated and/or analysed during the current study including a complete taxa list are available from the corresponding author on request.

**Acknowledgments:** We would like to thank numerous students who assisted with fieldwork, hydrochemical analyses in the laboratory and microscopic analyses of diatom samples.

**Conflicts of Interest:** The authors declare that they have no conflict of interest.

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
