# Peer review of "Diatom Red List Species Reveal High Conservation Value and Vulnerability of Mountain Lakes"

_diversity, doi:10.3390/d14050389_

Round 1
Reviewer 1 Report
PLEASE SEE ATTACHED FILE

Reviewer 2 Report
The revised manuscript is dedicated to the study of the mountain lake diatoms in connection with biodiversity conservation. The relevance of the research is beyond doubt. During the research, the authors used the standard methods of diatom research together with the estimation of morphological, physical, and chemical parameters and statistical analysis. The quality of the presentation of results is good enough. The Conclusions of the paper are adequate. The list of references contains the main sources of the research problem.
But I have some recommendations for the authors before the publication of the paper.
I understand that this is very time-consuming, but including a list of discovered species would make the MS more meaningful for a wide range of researchers.
Line 55: Start a sentence “Diatoms… “with a new paragraph, because this is a different semantic part
Line 67: You can delete “(hereafter called “RL”)”, since you mentioned it in Abstract.
Lines 105-108, 381, 405, 431, and further: Please, specify the authors of the taxa
Line 131: Mi=mixed better to correct to Mi=Mixed to make all abbreviations the same
Figure 2: Please, give the explanation of the scale from 1 to -1. What do blue and red colors mean? Some readers may not understand this
Figure 5: Please, explain the blue and red colors
Line 558: Please, write the authors' contributions, according to Diversity recommendation.

Reviewer 3 Report
Summary: The aim of the paper is work on the conservation value of mountain lakes (situated in the Alps) using the number, share and abundance of diatom Red List (RL) taxa and their relationship with environmental variables, diatom α and β diversity (uniqueness). Fourty-three lakes were retained leading to an important database, inducing relevant statistical conclusion. Planktonic, littoral and sediment diatoms were studied in each lake (when possible). It is an interesting paper with a robust methodology. The “discussion” is well documented. It is a very nice paper easy to read.
Specific comments:
Abstract: line15: modify where into when for “spatial descriptors were included wheN relevant”.
Materials and Methods:
2.1 Study site: modify in the 4 first lines “masl” into “m asl”.
Table 1: what means “ABT” and “also” AST”?
2.2: Sampling and lab procedures: line 184: suppress a space before [46].
- 7: “Environmental predictor variables were selected based on their variance inflation factors (VIF).”: how the variance was calculated?
Results
Page 8: what is Figure S2?
Page 9: in the text, Figure 4 appears before Figure 3. Please modify.
Page 10: line 264 suppress a space between “state.” And “After”.
Page 10: line 279: what is Figure S1 ?
Discussion:
P.15 line 399: put in lower case the “A” of “a group of lakes that is….”
P.15 line 482: put in lower case the “S” of “short-term interference…”.
